# Enhancing Professional Periodontal Therapy with a Novel PMA-Zeolite Application: A Clinical Study on Periodontal Outcomes and Microbiological Changes

**DOI:** 10.3390/jfb16080270

**Published:** 2025-07-22

**Authors:** Ines Đapić, Andrej Aurer, Jurica Žučko, Marinka Mravak-Stipetić, Marinka Baranović Baričević, Krešimir Pavelić, Fusun Ozer, Sandra Kraljević Pavelić

**Affiliations:** 1Department of Periodontology, University of Zagreb School of Dental Medicine, 10000 Zagreb, Croatia; idjapic@sfzg.hr (I.Đ.); aurer@sfzg.hr (A.A.); 2Faculty of Food Technology and Biotechnology, University of Zagreb, 10000 Zagreb, Croatia; jurica.zucko@pbf.unizg.hr; 3Faculty of Dental Medicine and Health, University Josip Juraj Strossmayer Osijek, 31000 Osijek, Croatia; mmstipetic@gmail.com (M.M.-S.); baricevicmarinka@gmail.com (M.B.B.); 4Medical Faculty, Juraj Dobrila University of Pula, 52000 Pula, Croatia; pavelic@unipu.hr; 5International Academy of Science and Arts in Bosnia and Herzegovina, Radnička cesta, 71000 Sarajevo, Bosnia and Herzegovina; 6School of Dental Medicine, University of Pennsylvania, Philadelphia, PA 19104, USA; ozerf@upenn.edu; 7Faculty of Health Studies, University of Rijeka, 51000 Rijeka, Croatia

**Keywords:** periodontitis, PMA-zeolite, professional prophylaxis, microbiota

## Abstract

Periodontitis is a chronic, multifactorial inflammatory disease characterized by the progressive destruction of the periodontal supporting tissues, including alveolar bone, potentially resulting in tooth loss. Etiopathogenesis involves a dysbiotic shift in the subgingival microbiota where the presence of pathogenic species such as *Porphyromonas gingivalis*, *Aggregatibacter actinomycetemcomitans*, and *Treponema denticola* has been documented. This disbalance is combined with an inadequate host immune response, often exacerbated by other systemic comorbidities including diabetes mellitus and cardiovascular diseases. Conventional therapy typically comprises mechanical debridement and adjunctive local or systemic antimicrobials, but emerging antibiotic resistance highlights a need for alternative adjuvant therapeutic strategies. The present descriptive analysis of microbiome and clinical trends study evaluated the adjuvant effects of a clinoptilolite-based zeolite material, namely PMA-zeolite, with professional prophylaxis on clinical and microbiological parameters in patients with chronic periodontitis over a 10-week period. Clinical assessment revealed significant reductions in bleeding on probing (BoP) and periodontal pocket depth (PD), indicating improved inflammatory status. Microbiome profiling demonstrated a marked decrease in key periodontal pathogens, suggesting that PMA-zeolite can help rebalance the oral microbiome. These findings suggest that the combined therapy exhibits promising anti-inflammatory and antimicrobial properties, indicating its role in promoting microbial homeostasis and reducing periodontal inflammation. However, further investigation through larger, controlled clinical trials is needed to validate the efficacy of the therapy.

## 1. Introduction

Periodontitis is a chronic inflammatory disease caused by microbial factors that affects the soft and hard tissues of the periodontium, potentially leading to tooth loss if untreated. Its progressive nature has prompted structured classification systems. According to the 2017 classification, periodontal diseases are grouped into gingivitis, periodontitis, and peri-implant diseases, with further subcategories based on severity and complexity [1,2,3,4]. Clinically, periodontitis is marked by attachment loss, alveolar bone resorption, and pocket formation with bleeding on probing. Its pathogenesis involves a dysregulated immune response to dysbiotic dental biofilms enriched with key pathogens, including *A. actinomycetemcomitans*, *P. gingivalis*, *T. denticola*, *T. forsythia*, *F. nucleatum*, and *Campylobacter* spp. This immune imbalance, worsened by systemic conditions such as diabetes and cardiovascular disease, underlies disease progression [5,6,7]. As Chitsazi et al. [8] noted, a key limitation of the 2017 classification is the unclear distinction between disease stages, leading to non-standardized treatment guidelines. Previous systems often paired antimicrobial therapy with mechanical debridement based on severity [9,10,11], while current approaches follow a stepwise strategy to control infection and inflammation [12,13]. Initial steps include supragingival biofilm removal and risk factor management, followed by subgingival instrumentation with the potential adjunctive use of local or systemic antimicrobials. If outcomes are insufficient, repeated instrumentation or surgical intervention may be required [14].

Systemic antibiotics, such as metronidazole, amoxicillin, ciprofloxacin, and azithromycin, are commonly used in periodontal therapy [15,16]. However, the use of local antibiotics has declined, partly due to the market withdrawal of many formulations for economic reasons [17]. Periodontitis presents a significant challenge in dental medicine, requiring innovative approaches beyond conventional treatments. As a result, zeolite-based materials have gained interest in periodontology and are currently being incorporated into resins, cements, irritants, and implant coatings given their antimicrobial and mechanical benefits [18]. In periodontology, zeolites have been proposed as antibacterial agents for the treatment of deep periodontal pockets, and zeolitic imidazolate frameworks (ZIFs) have been embedded in membranes for guided tissue or bone regeneration [19]. Zeolites are crystalline microporous aluminosilicates composed of a three-dimensional framework of corner-sharing SiO_4_ and AlO_4_ tetrahedra [20]. This arrangement forms the fundamental polymeric network structure. The aluminosilicate framework consists of a tetrahedral arrangement of silicon (Si^4^) and aluminum (Al^3^) cations, each coordinated with four oxygen anions (O^2^). These tetrahedral units function as the primary structural monomers of the zeolite framework [21]. The literature also makes specific connections between zeolites and other inorganic polymers. Research, specifically on geopolymers, highlights their relationship to zeolites, explaining that the synthesis of both materials typically requires high pH levels, concentrated alkaline solutions, atmospheric pressure, and moderate curing temperatures. Additionally, polymeric frameworks share structural and functional similarities with zeolites, especially in terms of their porous architecture and ion-exchange properties [22]. The crystalline aluminosilicate framework of zeolites leads to a well-organized network of microporous channels and cavities at the molecular scale. This characteristic architecture supports size-selective adsorption and ion-exchange processes, enabling targeted interactions with oral pathogens, pro-inflammatory mediators, and periodontal toxins via diverse therapeutic mechanisms [19].

Given the need for adjunctive therapies that effectively reduce inflammation, eliminate pathogenic bacteria, and prevent biofilm formation without adverse effects, this study aimed to evaluate the impact of a clinoptilolite-based material, namely, PMA-zeolite, on oral microbiota and periodontal inflammation. This descriptive analysis of microbiome and clinical trend study was performed in the context of an adjuvant therapy regimen combined with standard professional prophylaxis, with the goal of contributing to the understanding of the potential role of PMA-zeolite as an adjunct in the treatment of periodontitis. The novelty of the PMA-zeolite treatment was envisaged in the synergistic integration of PMA-zeolite with the standard prophylaxis. The PMA-zeolite adds effects that were ascribed to this material in previous studies and scientific literature. These are detoxification (heavy metal binding), as PMA-zeolite has increased surface area and adsorption capacity due to tribomechanical activation. It has the potential to bind bacterial toxins, heavy metals, and inflammatory mediators locally in the periodontal pockets. This provides a biologically detoxifying effect, which standard mechanical cleaning lacks. Moreover, anti-inflammatory effects have also been postulated for this material, potentially expected to reduce local inflammation. This may support tissue regeneration and healing in chronic periodontitis beyond what standard prophylaxis can achieve. At last, unlike antiseptics, PMA-zeolite does not indiscriminately kill bacteria and adsorbs pathogenic factors due to its physical-chemical properties in a selective manner without disrupting the entire oral microbiome. This is a property that was expected to promote long-term microbial balance. As a non-toxic material, PMA-zeolite avoids the side effects of chemical antiseptics like mucosal irritation.

## 2. Materials and Methods

### 2.1. Patient Population

The initial study design comprised 67 patients randomly assigned to the test group treated with PMA-zeolite and 22 patients assigned to the control group receiving only standard initial periodontal therapy, all diagnosed with periodontitis. The inclusion criteria were as follows: (1) patients diagnosed with stage II–IV periodontitis between 20 and 70 years old; (2) patients who agreed to participate in the study by signing dated informed consent, previously approved by the Ethics Committee, School of Dental Medicine University of Zagreb (13 September 2019. No: 05-PA-30-IX-9/2019); (3) patients do not use other local medications for treatment of periodontitis, such as antiseptic solutions or propolis; (4) patients willing to comply with the treatment plan and follow given oral hygiene instructions. The exclusion criteria were as follows: (1) patients who use other local oral therapy and those with complete dentures; (2) non-consented patients and those unable to participate due to any personal or objective reason.

### 2.2. Study Design

After the initial evaluation, 31 patients dropped out due to various reasons including the earthquake in Zagreb and the COVID-19 pandemic (Appendix A). There were 36 patients assigned to the PMA-zeolite treatment group, and 6 patients were assigned as controls not receiving PMA-zeolite treatment, yielding 42 patients in total. All subjects were recruited and treated at the Department of Periodontology, School of Dental Medicine University of Zagreb, from February 2019 to 7 July 2021. Diagnosis of periodontitis (stage II–IV) was established in all recruited study subjects by experienced periodontist clinicians, according to the latest classification of periodontal diseases from 2017.

Baseline medical and dental histories were collected, and the following periodontal indices were recorded: probing depth (PD), papilla bleeding index (PBI), and approximate plaque index (API). These measurements were taken at the start of the study and again 12 weeks post-treatment during the re-evaluation appointment. Patients in the test group (n = 36) used PMA-zeolite containing toothpaste for 12 weeks, and microbiological samples were taken at the baseline and after 12 weeks (follow-up) (Figure 1).

At the baseline, the subjects were given detailed instructions to brush their teeth with a medium toothbrush twice a day for 3–5 min using a modified Bass brushing technique and interdental toothbrushes once a day. Test subjects were given PMA-zeolite toothpaste (toothpaste with the addition of 5–10% PMA-zeolite provided by Panaceo International GmbH, Austria). Afterwards, subgingival scaling and root planning were performed under local anesthesia [20]. Additionally, subgingival irrigation was performed in the test subjects using freshly prepared PMA-zeolite solution containing 500 mL of demineralized water and 5 g of PMA-zeolite powder (clinoptilolite-based material obtained from Panaceo International GmbH, Austria). The procedure is termed as PZ+SP (PMA-zeolite plus Standard Prophylaxis) in the manuscript. The patients were instructed to return for subgingival irrigation after 1 week, when this procedure was repeated. According to the recent literature, a minimum follow-up to evaluate the efficacy of nonsurgical periodontal therapy (NSPT) in treating periodontitis in patients with concurrent systemic conditions is 3 months. This is also the time for recall; however, evidence for a specific recall interval (e.g., every 3 months) for all patients following periodontal therapy depends on the various factors such as risk factors, type of periodontal disease, and underlying systemic disease, and is preferred to be individually determined. In our study, we cited the study of Kaner D et al. [10] which indicated that administration of antimicrobial therapy immediately after initial mechanical debridement provides better clinical findings than late administration of systemic antimicrobial therapy with mechanical re-instrumentation after 3 months. In our cohort, based on this knowledge regarding the influence of time on the outcome of therapy, we determined a three-month treatment period and proved the effectiveness of the PMA-Zeolite therapy for 10 weeks on the decreased gingival inflammation, decreased bleeding on probing (BoP), and pocket depth status (PD) parameters. We consider the period of monitoring the effect of zeolite therapy on periodontal changes to be optimal for 12 weeks in order to observe, assess, and measure the therapeutic effect.

In all patients, clinical measurements were recorded. Microbiological samples were obtained from the four deepest pockets using paper points and from sublingual saliva using microbiology brushes. These samples were pooled together for each patient for the purposes of the microbiome analysis. After 12 weeks, microbiological samples from periodontal pockets and saliva were collected again at the follow-up. The patients were questioned about the influence of periodontal therapy on self-reported oral health, with patients rating the treatment benefits as better, worse, or the same as before.

### 2.3. Materials and Procedures for Wide Genome Sequencing (WGS)

For total microbiome analysis and sequencing, salivary and periodontal pocket samples were pooled at the baseline and follow-up in small sterile polypropylene containers (Eppendorf Safe-Lock Tubes 1.5 mL, PCR clean, Eppendorf, Nijmegen, Netherlands), and paper points were used for periodontal pocket sampling (Dia-ProT Paper Points, DiaDent Group International, DiaDent Europe B.V., Almere, The Netherland). In addition, the study also used microbiological brushes (Rambrush, RI.MOS s.r.l, Mirandola (MO), Italy), small containers for liquid nitrogen, and liquid nitrogen for sample transportation. Samples were stored in standard PBS solution at −80 °C until DNA extraction. For DNA extraction, the Chemagic 360 system was used (Geneplanet, Ljubljana, Slovenia). Next-generation shotgun sequencing of the whole oral metagenome was performed using the Novaseq 6000 platform (Illumina Inc., San Diego, CA, USA) under the following conditions: read length: pair-end 150 bp; library preparation kit: NEB Next^®^ Ultra™ DNA Library Prep Kit (New England Biolabs (NEB), Amersfoort, The Netherlands); >2 Gb data/sample. The in-house and Kraken metagenomic sequence classification was used for identification.

### 2.4. Statistical Analysis

The clinical and oral microbiomes of patients with periodontitis before and after therapy were compared for a change in clinical findings and bacterial diversity. All data were statistically analyzed, including periodontal clinical indices (PD, PBI, API) and genomic analysis of the whole oral microbiome based on sequencing data from the microbiological samples. Patients were given a questionnaire at the follow-up (12 weeks from baseline) to assess their perception of therapeutic success. For the assessment of clinical parameters in patients who completed the PZ+SP therapy, the following tests were performed for BoP and PD: Fisher’s exact tests for count data and McNemar’s chi-squared tests with continuity correction.

Whole oral metagenome shotgun reads were mapped onto the reference human genome (hg37) using the bwa 0.7.17 aligner. Reads not mapped onto the reference human genome were considered of microbial origin and were used to determine taxonomy with the MetaPhlAn 3 program. Here, mpa_v30_CHOCOPhlAn_201901 was used as a database of marker genes for taxonomic profiling. Unmapped reads were used as paired-end reads natively by MetaPhlAn (without joining them beforehand). MetaPhlAn taxonomic profiles of individual samples were merged using merge_metaphlan_tables.py script into a single taxonomic table. STAMP v 2.1.3. was used to determine statistical differences in the obtained taxonomic profiles. Raw data are deposited at EBI’s ENA database under project accession PRJEB88546, accessible at https://www.ebi.ac.uk/ena/browser/view/PRJEB88546 (accessed on 18 April 2025).

## 3. Results

Clinical improvement was observed in all patients following the initial periodontal therapy [23] and 10 weeks of PZ+SP treatment. This response included a reduction in probing pocket depth (PD) and an improvement in gingival health, with the gingiva color transitioning from red to pink, indicating a healthy state. A noticeable decrease in inflammation, swelling, and gingival sensitivity, as well as in the surrounding mucosa, was observed in all patients. After a 12-week follow-up, clinical examination revealed further improvements in periodontal health, including reduced bleeding, lower PD values, and a decrease in the approximate plaque index (API). A subjective assessment of the oral cavity condition was given in oral form by each patient. Patients were asked the following question: “How do you feel related to the condition in your oral cavity?” The following answers were given: “I feel better”—24 patients; “I feel worse”—7 patients; “I feel the same”—5 patients.

Statistical analysis was performed for clinical parameters for all patients who successfully completed the PZ+SP treatment cycle (Table 1). Two parameters were analyzed, namely, the bleeding on probing (BoP) and pocket depth (PD) parameters. BoP is a test/index that shows the degree of inflammation followed by bleeding. It also serves as a motivational index, as reduced gingival bleeding after periodontal therapy or during brushing encourages patients to remain committed to treatment. PD values up to 3 mm are considered healthy, 4–5 mm shallow, and 6 mm or greater are considered deep pockets. The approximal plaque index (API) was not assessed as a clear indicator of the overall combined prophylaxis and adjuvant PZ+SP treatment, as it correlates directly with the standard professional periodontal prophylaxis received at study baseline.

Here, chi-squared tests were used to compare the following groups: controls (K) vs. PMA therapy (N) and PMA therapy_start (N_S) vs. PMA therapy_end (N_E), where N stands for PMA-therapy (Table 1). The results at the end of PZ+SP therapy showed a marked statistical difference towards improved values for BoP (Fisher’s exact test for count data, *p*-value = 1.187 × 10^−9^; McNemar’s chi-squared test with continuity correction = 21.043, df = 1, *p*-value = 4.49 × 10^−6^) and PD (Fisher’s exact test for count data, *p*-value = 4.684 × 10^−8^; McNemar’s chi-squared test with continuity correction = 14.062, df = 1, *p*-value = 0.0001768) compared to the beginning of the study before PZ+SP treatment. Obtained results are comparable with results from a recent large meta-analysis showing that standard non-surgical prophylaxis may reduce mean PD and BoP in patients with concurrent systemic conditions, but only after 3 months compared to the control group [24]. Thus, adjuvant PZ+SP treatment improves the patient’s clinical status.

The sequencing results for the patients with available sequencing data were correlated with two clinical parameters, namely the BoP and PD (Table 2). Comparisons between the following groups were performed using chi-squared tests: controls (K) vs. PMA therapy (N) and PMA therapy_start (N_S) vs. PMA therapy_end (N_E) (Table 2). The groups were assessed based on the BoP (groups 1–4). None of the group comparisons in patients with available sequencing data were statistically significant for the BoP parameter (*p* = 0.65 for K vs. N; *p* = 0.39 for N_S vs. N_E). For PD testing, subjects were grouped into three groups based on the determined depth of the pockets: subjects with deep pockets, shallow pockets, or both (ALL). Statistical testing for differences between therapy start (N_S) and therapy end (N_E) in patients with sequencing data did not show significant differences between the groups (*p* = 0.14 for N_S vs. N_E). The sequencing results herein are presented only for the portion of patients where extraction of total metagenomic DNA was adequate for the requirements of the sequencing (N = 7). In all patients with diagnosed periodontitis, specifically enriched phyla included *Proteobacteria*, *Firmicutes*, *Bacterioidetes*, *Actinobacteria*, *Synergistetes*, *Spirochaetes*, *Fusobacteria*, and *Chloroflexi* (Welch’s t-test, two-sided, *p* < 0.05) (Figure 2).

Based on phylogenetic analysis of 16S rRNA gene, the *Proteobacteria* phylum is divided into six classes (previously regarded as subclasses of the phylum): *Alphaproteobacteria*, *Betaproteobacteria*, *Gammaproteobacteria*, *Deltaproteobacteria*, *Epsilonproteobacteria*, and *Zetaproteobacteria* [25]. Individual statistically relevant class differences were obtained for seven patients (two-sided Fisher’s exact test with Benjamini–Hochberg FDR correction, *p* < 0.05) who showed a highly individualized response in class comparison. This finding may be due to the large oral microbiome variations already documented in the literature [26]. These individual data are presented in the Appendix A. Significant results from a general comparison of the microbiome status of all tested microbiological samples before and after therapy are presented in Figure 3, Figure 4 and Figure 5.

## 4. Discussion

Therapeutic approaches to the treatment of periodontitis vary depending on the stage of the disease and may range from improving oral hygiene and removing bacterial biofilm with mechanical instrumentation of the root surface to adjunctive local or systemic antimicrobial therapy, and even surgical treatment. Initial periodontal therapy typically includes thorough oral hygiene instructions, debridement of the supragingival and subgingival biofilm, control of proven risk factors of periodontal diseases like smoking or diabetes, and, where appropriate, the use of local or systemic antimicrobial or anti-inflammatory medication [27]. In advanced or aggressive cases of periodontal disease, systemic antibiotics are employed as adjunctive therapy to mechanical debridement, preferably as part of nonsurgical periodontal therapy [13,28]. However, their frequent use increases antimicrobial resistance [17,29].

To avoid the limitations of traditional treatment, especially regarding mechanical instrumentation as well as the application of local and systemic antibiotics and surgical interventions, some alternative therapeutic approaches were suggested, including the application of photodynamic antimicrobial therapy [30], the application of natural substances such as polyphenols [31], and the use of probiotics as an adjunct to periodontal therapy. These methods may restore microbiota balance, reduce inflammation, and improve the clinical outcomes in periodontitis [32]. Antimicrobial photodynamic therapy (aPDT) was found to be more effective than the administration of systemic antibiotics [18] and provided additional clinical improvements in the treatment of residual periodontal pockets, as shown by Xue et al. [32]. Although some additional auxiliary treatment modalities were proposed, including prebiotics/synbiotics, statins, pro-resolving mediators, omega-6 and -3, ozone, and epigenetic therapy, in addition to the traditional periodontal therapy, these methods did not show a significant clinical effect except as an aid to improve the outcomes of non-surgical periodontal therapy [14].

Recent studies have shown that zeolite-based materials are effective in preventing biofilm formation across various applications. For instance, ZnO/ZeoNC nanocomposites have inhibited biofilms of both standard and clinical strains of *Staphylococcus aureus* [33]. Similarly, silver-impregnated zeolite-clay granules and porous pot filtration systems have demonstrated antimicrobial activity [34,35], as has silver-ion-exchanged zeolite A, particularly against biofilm formation by *Vibrio* spp. marine pathogens [36]. The synthetic allumosilicate zeolite 4A also interferes with bacterial communication and disrupts biofilm formation in *Pseudomonas aeruginosa*-related infections [37]. The natural allumosilicate zeolite clinoptilolite strongly inhibits biofilm formation and twitching motility of bacteria due to immobilization of bacterial cells onto zeolite particles [38]. With this in mind, the results obtained in the present study were partially expected. PZ+SP treatment has indeed improved clinical periodontal indices in all patients after 12 weeks of PZ+SP treatment. The clinical results at the end of PZ+SP treatment showed a statistical difference toward improved values for the BoP (Fisher’s exact test for count data, *p*-value = 1.187 × 10^−9^; McNemar’s chi-squared test with continuity correction = 21.043, df = 1, *p*-value = 4.49 × 10^−6^) and PD (Fisher’s exact test for count data, *p*-value = 4.684 × 10^−8^; McNemar’s Chi-squared test with continuity correction = 14.062, df = 1, *p*-value = 0.0001768).

The improved clinical state and subjective feeling were also confirmed at the follow-up. To date, one meta-analysis has evaluated the effectiveness of standard non-surgical prophylaxis in reducing mean PD and BoP in periodontic patients under concurrent systemic conditions. The study showed the effectiveness of the procedure in alleviating the symptoms, but only after 3 months, compared to the control group [24]. PZ+SP treatment for 10 weeks has the potential to improve clinical status.

The sequencing results of the current study complemented the clinical findings, revealing an oral microbiome composition that closely matched previously documented profiles of both healthy individuals [39] and those with periodontitis [40,41,42]. Among all patients diagnosed with periodontitis, the microbiome analysis consistently showed an increased presence of several bacterial phyla, including *Proteobacteria*, *Firmicutes*, *Bacteroidetes*, *Actinobacteria*, *Synergistetes*, *Spirochaetes*, *Fusobacteria*, and *Chloroflexi* (Figure 1). Results from the Human Microbiome Project showed that the main genus among intraoral sites with greater than 10% abundance and present in more than 75% of samples was *Streptococcus*. Other core members with greater than 1% abundance across at least 80% of samples from one or more sites included those bacteria from the family *Pasteurellaceae*; those from the genera *Gemella*, *Veillonella*, *Prevotella*, *Fusobacterium*, *Porphyromonas*, *Neisseria*, *Capnocytophaga*, *Corynebacterium*, and *Actinomyces*; and those from the orders *Lactobacillales* and *Lachnospiraceae* [43].

The precise composition of the oral microbiome is inherently variable and resists standardization due to a multitude of dynamic factors that continuously shape microbial populations, including diet, oral hygiene habits, circadian rhythms, and inter-individual differences [44,45]. The identified enriched phylum in patients with periodontitis in the present study included *Proteobacteria*, *Firmicutes*, *Bacterioidetes*, *Actinobacteria*, *Synergistetes*, *Spirochaetes*, *Fusobacteria*, and *Chlorofelxi*. This finding is in line with previously established information on the human oral microbiome and literature data on periodontitis patients. Indeed, the Human Oral Microbiome Database (HOMD, www.homd.org) includes 619 taxa in 13 phyla: *Actinobacteria*, *Bacteroidetes*, *Chlamydiae*, *Chloroflexi*, *Euryarchaeota*, *Firmicutes*, *Fusobacteria*, *Proteobacteria*, *Spirochaetes*, *SR1*, *Synergistetes*, *Tenericutes*, and *TM7.* Moreover, the following phyla are correlated with the symptoms of periodontitis: *Spirochaetes*, *Synergistetes*, and *Bacteroidetes* [46].

The statistically significant microbiome analysis comparing all samples before and after PZ+SP treatment revealed specific changes at the class, family, genus, and species level as summarized in Table 3. In particular, PZ+SP treatment decreased the abundance of the class *Deltaproteobacteria* that are found to be increased in disease and have been identified by some authors as a possible marker of microbiota instability, thus predisposing individuals to disease onset [26,47]. Moreover, PZ+SP treatment decreased the abundance of the *Desulfobulbaceae* family that plays a role in inflammation within the oral cavity of patients with periodontitis [48]. This effect may be attributed to the adsorption of sulfur compounds by PMA-zeolite which may lead to decreased levels of these pro-inflammatory bacterial metabolites that support the growth of these taxa. Accordingly, a shift in the microbial community toward a more balanced state may then occur.

PZ+SP zeolite treatment also increased the abundance of the *Lachnospiraceae* family, which has dual roles in various pathologies. In the context of this treatment, the increase in *Lachnospiraceae* may serve as a potential marker of reduced histamine content in the oral cavity [49,50,51,52]. *Lachnospiraceae*’s potential to modulate histamine levels or maybe outcompete histamine-producing taxa may contribute to a less inflammatory environment. PZ+SP treatment decreased the abundance of the genus *Desulfobulbus*, which plays a role in inflammation within the oral cavity of patients with periodontitis [53,54,55,56,57]. *Desulfobulbus oralis* has previously been found to increase levels of IL-1α and IL-6 that activate the pro-inflammatory pathways (57). Both taxa may be suppressed by PMA-zeolite which leads to diminished inflammatory modulators. PZ+SP treatment increased the abundance of the genus *Oribacterium* [58] and decreased the abundance of the species *Desulfobulbus oralis*, an established oral pathobiont that can trigger a pro-inflammatory response in oral epithelial cells, suggesting its direct role in the development of periodontal disease [58,59,60]. In addition, PZ+SP treatment decreased the abundance of the species *Gemella bergeri* that can become pathogenic and cause infections in patients with poor oral hygiene [61]. This reduction may result from the enhanced adsorptive properties of PMA-zeolite, which can adsorb bacterial metabolites or even disrupt biofilm formation. This mechanism may impair the ecological niche for opportunistic pathogens like *G. bergeri*^.^ At last, PZ+SP treatment increased the abundance of the species *Oribacterium* sp. To date, this species has not been associated with periodontitis and is mainly studied in the context of other oral diseases [62,63]. From the obtained results, it may be deduced that PZ+SP has a modulatory effect on the oral microbiome through selective suppression of inflammation-associated taxa, i.e., *Deltaproteobacteria*, *Desulfobulbaceae*, *Desulfobulbus oralis*, and *Gemella bergeri*. These taxa have already been described in the literature as oral dysbiosis markers. The treatment also increased the abundance of genera *Oribacterium* and members of *Lachnospiraceae* which points to a shift toward a more anti-inflammatory microbial profile. These observations may be explained by the PMA-Zeolite’s ability to adsorb pro-inflammatory metabolites and interfere with biofilm formation while at the same time supporting microbial homeostasis.

A major limitation of this study, however, was the incomplete follow-up of control patients who received only conventional periodontal treatment, as these individuals did not return for their 10-week evaluation. The absence of this control data presents a gap in the analysis. The pattern of dropout among these patients suggests that a lack of perceived therapeutic benefit may have contributed to their decision to discontinue participation.

## 5. Conclusions

This descriptive analysis of microbiome and clinical trend study demonstrates that 10 weeks of treatment with PMA-zeolite, when used as an adjunct to professional prophylaxis, offers significant clinical and microbiological benefits for the patients in the non-surgical treatment of periodontitis. The adjunctive application of PMA-zeolite resulted in a reduction in mucosal redness and clinical inflammation, as reflected by decreased bleeding on probing (BoP), and a significant improvement in periodontal pocket depth (PD). These findings suggest that PMA-zeolite accelerates clinical improvements compared to standard prophylaxis alone. Given the increasing concerns about antibiotic resistance, PMA-zeolite presents a promising non-antibiotic alternative or complementary treatment strategy in periodontal care.

Oral professional plaque removal combined with PMA-zeolite treatment also reduced some microbial markers linked with oral dysbiosis-inflammation, including *Deltaproteobacteria*, *Desulfobulbaceae*, *Desulfobulbus*, *Desulfobulbus oralis*, and *Gemella bergeri*, while favoring the growth of *Oribacterium* along with *Lachnospiraceae*, which are associated with reduced histamine-related inflammation. These findings contribute to current knowledge by demonstrating that PMA-zeolite can influence microbial composition in a way that supports periodontal health, through mechanisms including the adsorption of bacterial toxins, pro-inflammatory metabolites, and interference with pathogenic biofilm formation. The observed microbial shifts suggest that PMA-zeolite promotes rebalancing of the oral microbiome, contributing to reduced inflammation and improved clinical outcomes.

These findings support the potential of PMA-zeolite as an effective adjunct that can improve the outcomes of conventional nonsurgical periodontal therapy in patients with periodontitis. However, the efficacy of PMA-zeolite must be confirmed through larger, long-term clinical trials involving broader patient populations. This study offers preliminary insight into its molecular effects but remains largely descriptive. Future research needs to explore its impact on cytokine signaling, immune regulation, and biofilm dynamics to clarify its full therapeutic potential.

## Figures and Tables

**Figure 1 jfb-16-00270-f001:**
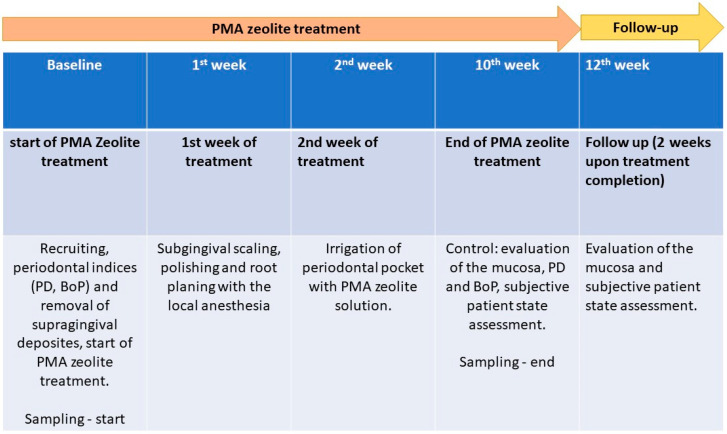
Summary of the study course and PZ+SP treatment plan.

**Figure 2 jfb-16-00270-f002:**
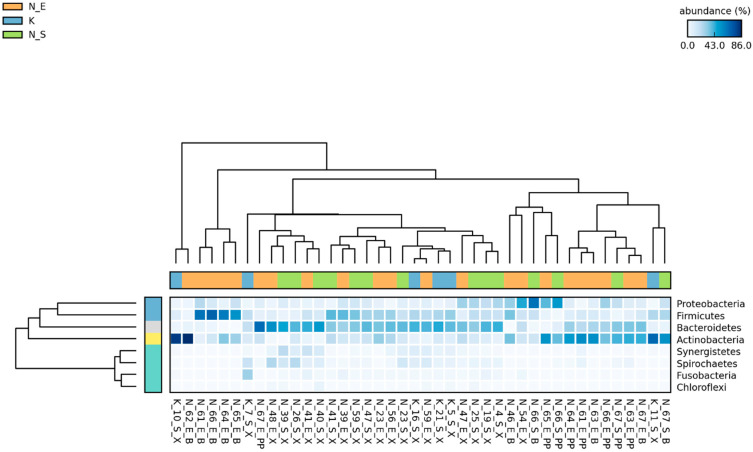
Enriched phyla in all patients diagnosed with chronic periodontitis as assessed using whole-genome sequencing included *Proteobacteria*, *Firmicutes*, *Bacterioidetes*, *Actinobacteria*, *Synergistetes*, *Spirochaetes*, *Fusobacteria*, and *Chlorofelxi* (ANOVA, Tukey–Kramer, eta squared, *p* < 0.05).

**Figure 3 jfb-16-00270-f003:**
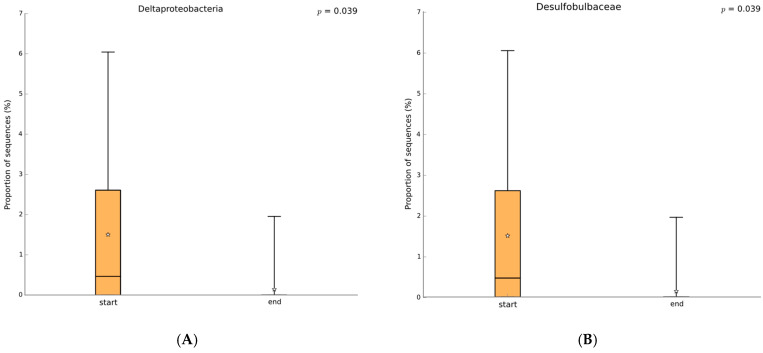
Class-level comparison of the microbiome status in samples from ALL patients before and after therapy (Welch’s test, two-sided, *p* < 0.05 with no correction, orange before therapy, blue after therapy). (**A**): *Deltaproteobacteria*; (**B**): *Desulfobulbaceae*; (**C**): *Lachnospiraceae*. The box-and-whiskers graphics show the median of the data as a line, the mean of the data as a star, the 25th and 75th percentiles of the data as the top and bottom of the box, and uses whiskers to indicate the most extreme data point within 1.5*(75th–25th percentile) of the median. Data points outside of the whiskers are shown as crosses (+). The sign of two crosses ‡ stays for two samples.

**Figure 4 jfb-16-00270-f004:**
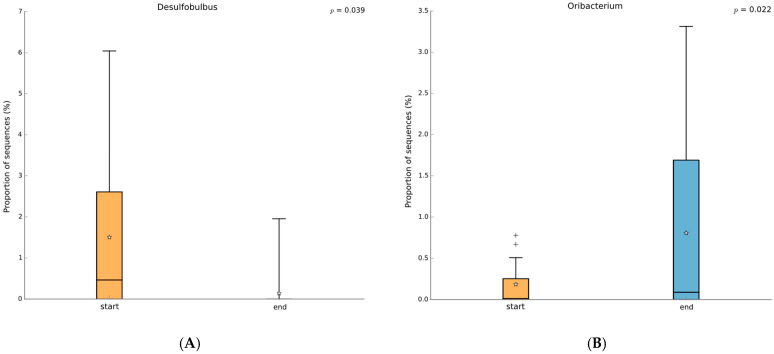
Genus-level comparison of the microbiome status in samples from ALL patients before and after therapy (Welch’s test, two-sided, *p* < 0.05 with no correction, orange before therapy, blue after therapy). (**A**): *Desulfobulbus*; (**B**): *Oribacterium*. The box-and-whiskers graphics show the median of the data as a line, the mean of the data as a star, the 25th and 75th percentiles of the data as the top and bottom of the box, and uses whiskers to indicate the most extreme data point within 1.5*(75th–25th percentile) of the median. Data points outside of the whiskers are shown as crosses (+).

**Figure 5 jfb-16-00270-f005:**
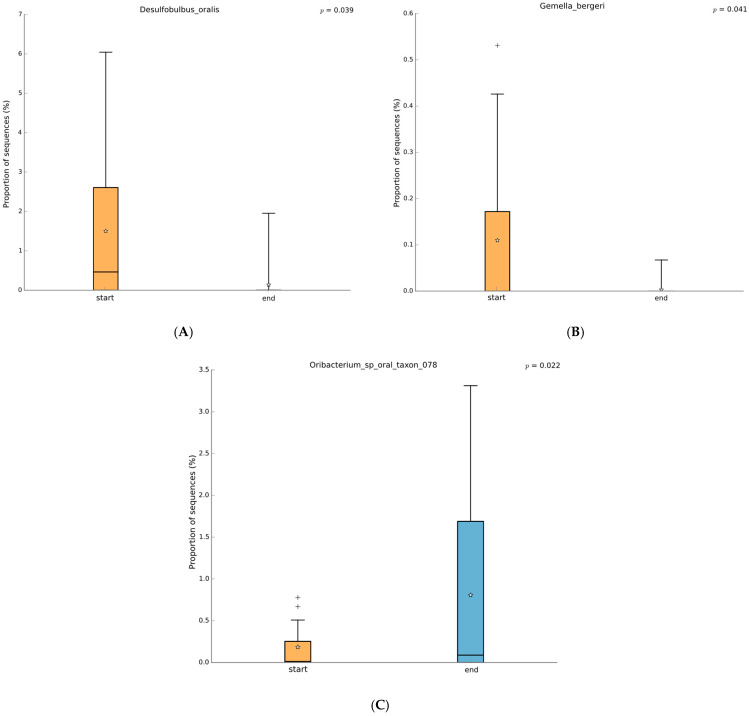
Species-level comparison of the microbiome status of samples from ALL patients before and after therapy (Welch’s test, two-sided, *p* < 0.05 with no correction, orange before therapy, blue after therapy). (**A**): *Desulfobulbus oralis*; (**B**): *Gemella bergeri*; (**C**): *Oribacterium* sp. The box-and-whiskers graphics show the median of the data as a line, the mean of the data as a star, the 25th and 75th percentiles of the data as the top and bottom of the box, and uses whiskers to indicate the most extreme data point within 1.5*(75th–25th percentile) of the median. Data points outside of the whiskers are shown as crosses (+).

**Table 1 jfb-16-00270-t001:** Clinical parameters assessed in patients (N = 36) at baseline and the end of therapy. Control patients (N = 6) who received only standard periodontic prophylaxis are presented at the end of the table separately. Categories for pocket depth statistical analysis were assessed as follows: shallow + deep = DEEP; healthy + shallow = SHALLOW; healthy + shallow + deep = ALL. Patients were categorized in the following groups: N_S, start of the PZ+SP treatment; N_E, end of the PZ+SP treatment.

Patient No.	BoP_N_S	BoP_N_E	API_N_S	API_N_E	PD_N_S	PD_N_E
PZ+SP TREATMENT GROUP						
1	3, 4	2, 3	+	+/−	>3	<3
2	3, 4	3, 4	+	-	>3	<3
3	3, 4	2, 3	+	-	>3	<3
4	3, 4	2, 3	+	-	>3	<3
5	2, 3, 4	3	+	+/−	>3	3
6	3, 4	3	+	-	>3	<3
7	2, 3, 4	2, 3	+	-	>3	2, 3
8	3, 4	2, 3	+	-	>3	2, 3
9	3, 4	2, 3	+	-	>3	<3
10	3, 4	3	+	-	>3	3, 4
11	3, 4	3	+	-	>3	5, 6
12	3, 4	2, 3	+	-	>3	3, 4
13	3, 4	2, 3	+	+/−	>3	2, 3
14	3, 4	3	+	-	>3	<3
15	3, 4	3	+	-	>3	<3
16	3, 4	3	+	-	>3	<3
17	3, 4	3, 4	+	-	>3	<3
18	3, 4	3	+	-	>3	<3
19	3, 4	3	+	-	>3	<3
20	3, 4	3	+	-	>3	3, 4, 5
21	3, 4	2, 3	+	-	>3	4, 5
22	3, 4	2, 3	+	-	>3	3, 4
23	3, 4	2, 3	+	-	>3	3, 4
24	3, 4	3, 4	+	-	>3	<3
25	3, 4	2	+	+/−	>3	<3
26	3, 4	2	+	+/−	>3	<3
27	3, 4	2, 3	+	-	>3	<3
28	3, 4	2, 3	+	-	>3	2, 3, 4
29	3, 4	2, 3	+	-	>3	3, 4
30	3, 4	2, 3	+	-	>3	2, 3, 4
31	3, 4	2, 3	+	-	>3	3
32	3, 4	2, 3	+	-	>3	3
33	3, 4	2	+	-	>3	3, 4
34	3, 4	2, 3	+	-	>3	2, 3
35	3, 4	2, 3	+	-	>3	3, 4
36	3, 4	2, 3	+	-	>3	3, 4
CONTROL GROUP						
1	3, 4	*	+	*	>3	*
2	3, 4	*	+	*	>3	*
3	3, 4	*	+	*	>3	*
4	3, 4	*	+	*	>3	*
5	3, 4	*	+	*	>3	*
6	3, 4	*	+	*	>3	*

* Control patients who received only standard periodontic prophylaxis did not return for follow-up at the 10-week period from the beginning of the study. Thus, these data are unavailable. Such dropouts are common due to a lack of disease improvement (lack of treatment effect) that increases fear of continued therapy.

**Table 2 jfb-16-00270-t002:** Clinical parameters assessed in patients with sequencing data. Categories for pocket depth statistical analysis were assessed as follows: shallow + deep = DEEP; healthy + shallow = SHALLOW; healthy + shallow + deep = ALL. Patients were categorized into the following groups: K, control; N, PZ+SP treatment; N_S, start of the PZ+SP treatment; N_E, end of the PZ+SP treatment.

No.	ID	BoP	Bleeding	Group	Timepoint	Deepness of the Pocket
1	N_48_E_X	3, 4	high	N	N_E	DEEP
2	N_40_S_X	3, 4	high	N	N_S	DEEP
3	N_59_E_X	3, 4	high	N	N_E	DEEP
4	N_41_S_X	3, 4	high	N	N_S	SHALLOW
5	N_47_E_X	3, 4	high	N	N_E	DEEP
6	N_59_S_X	3, 4	high	N	N_S	DEEP
7	N_39_S_X	3, 4	high	N	N_S	DEEP
8	N_56_E_X	3, 4	high	N	N_E	DEEP
9	N_23_S_X	1, 2, 3, 4	full	N	N_S	DEEP
10	N_25_S_X	3, 4	high	N	N_S	DEEP
11	N_4_S_X	2	low	N	N_S	ALL
12	N_19_S_X	1, 2, 3, 4	full	N	N_S	ALL
13	N_26_S_X	2, 3, 4	high	N	N_S	ALL
14	N_41_E_X	3, 4	high	N	N_E	SHALLOW
15	N_47_S_X	3, 4	high	N	N_S	DEEP
16	N_39_E_X	3, 4	high	N	N_E	DEEP
17	N_54_E_X	3, 4	high	N	N_E	SHALLOW
18	N_23_E_X	1, 2, 3, 4	full	N	N_E	DEEP
19	N_67_E_B	3, 4	high	N	N_E	ALL
20	N_66_E_PP	3, 4	high	N	N_E	ALL
21	N_46_E_B	2, 3	avg	N	N_E	DEEP
22	N_66_S_PP	3, 4	high	N	N_S	ALL
23	N_67_S_PP	3, 4	high	N	N_S	ALL
24	N_65_E_PP	3, 4	high	N	N_E	SHALLOW
25	N_66_S_B	3, 4	high	N	N_S	ALL
26	N_61_E_PP	2, 3	avg	N	N_E	SHALLOW
27	N_67_S_B	3, 4	high	N	N_S	ALL
28	N_67_E_PP	3, 4	high	N	N_E	ALL
29	N_63_E_B	3, 4	high	N	N_E	ALL
30	N_66_E_B	3, 4	high	N	N_E	ALL
31	N_65_E_B	3, 4	high	N	N_E	SHALLOW
32	N_64_E_B	3, 4	high	N	N_E	SHALLOW
33	N_63_E_PP	3, 4	high	N	N_E	ALL
34	N_64_E_PP	3, 4	high	N	N_E	SHALLOW
35	N_62_E_B	3, 4	high	N	N_E	ALL
36	N_61_E_B	2, 3	avg	N	N_E	SHALLOW
37	K_11_S_X	3, 4	high	K	K	
38	K_10_S_X	3, 4	high	K	K	
39	K_5_S_X	3, 4	high	K	K	
40	K_7_S_X	3, 4	high	K	K	
41	K_21_S_X	3, 4	high	K	K	
42	K_16_S_X	3, 4	high	K	K	

**Table 3 jfb-16-00270-t003:** Significantly relevant results (Welch’s test, two-sided, *p* < 0.05 with no correction) of the oral microbiota changes in periodontic patients after 10 weeks of PZ+SP.

Taxonomy Level	Oral Microbiome Representative	Direction of Change in Abundance Upon PZ+SP Treatment
Class	*Deltaproteobacteria **	**↓**
Family	*Desulfobulbaceae **	**↓**
Family	*Lachnospiraceae **	**↑**
Genus	*Desulfobulbus **	**↓**
Genus	*Oribacterium*	**↑**
Species	*Desulfobulbus oralis **	**↓**
Species	*Gemella bergeri **	**↓**
Species	*Oribacterium* sp.	**↑**

* Established indicators of inflammation and oral dysbiosis in oral disease.

## Data Availability

The original data presented in the study are openly available in EBI’s ENA database under project accession PRJEB88546, accessible at https://www.ebi.ac.uk/ena/browser/view/PRJEB88546 (accessed on 1 May 2025).

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
