# Peer review of "Enhancing Professional Periodontal Therapy with a Novel PMA-Zeolite Application: A Clinical Study on Periodontal Outcomes and Microbiological Changes"

_jfb, 2025, doi:10.3390/jfb16080270_

Round 1

Reviewer 1 Report

Comments and Suggestions for Authors

This article demonstrates a high level of research design, methodology, data analysis, and result interpretation, providing valuable insights for the treatment of periodontitis. A new type of PMA Zeolite material was studied for its auxiliary effect in the treatment of periodontitis, providing new ideas and methods for the treatment of periodontitis and demonstrating a certain degree of innovation. Although there are some limitations, the research results have certain clinical significance and application prospects, and the following modifications should be completed before acceptance.

  1. The font size of some data graphs is too small, which makes it difficult for readers to quickly capture useful information. It is recommended to make modifications.
  2. There are some language expressions in the text that are not accurate or fluent enough, such as complex sentence structures that may make it difficult for readers to understand. It is recommended to further polish them.
  3. It is recommended to add references to the latest related literature, such as, Corrosion Communications 17 (2025) 35–43; Biomaterials 318 (2025) 122992.

Author Response

Thank you for the positive reviews. The graphs cannot be modified as they have been generated by the software. We already tried to adjust the font directly but this option is not available and we decided not to modify the raw data figure files. We hope you will agree with us on this point.

The paper has undergone a professional language revision performed by the MDPI office.

The suggested references cover the topic of alloys and we do not understand exactly where to add them. For now, we suggest not to add them but are open the reviewer suggestion on the exact position in the paper if needed.

Reviewer 2 Report

Comments and Suggestions for Authors

This paper reports a clinical study investigating the adjunctive use of PMA-Zeolite, a clinoptilolite-based material, in conjunction with professional mechanical periodontal therapy, evaluating its effects on clinical outcomes and microbiological profiles in patients with chronic periodontitis. The topic is relevant and timely, particularly in light of increasing antimicrobial resistance and the search for alternative therapeutic strategies. The integration of a functional biomaterial with clinical periodontal therapy aligns with the scope of the Journal of Functional Biomaterials and could be considered for publication following revisions:
1. Clarify the novelty of PMA-Zeolite treatment compared to existing zeolite-based strategies—what makes this formulation or approach distinct?
2. The repeated phrase “PMA-Zeolite treatment combined with standard prophylaxis” affects readability; consider using a consistent abbreviation after first mention.
3. Strengthen the justification for the 12-week treatment period—why was this specific duration chosen, and how does it compare to similar studies?
4. The discussion of microbiome shifts is descriptive but could benefit from deeper interpretation—how do the observed taxonomic changes translate into clinical or mechanistic insights?
5. The conclusion is overly concise; it should be expanded to highlight how the findings contribute to current knowledge, propose mechanisms, and outline specific directions for future research.

Comments on the Quality of English Language

The English Quality can be improved.

Author Response

  1. Clarify the novelty of PMA-Zeolite treatment compared to existing zeolite-based strategies—what makes this formulation or approach distinct

The novelty of the PMA-Zeolite treatment lies in the synergistic integration of PMA zeolite with the standard prophylaxis. The PMA zeolite adds effects that were ascribed to this material in previous studies and scientific literature. These are, detoxification (heavy metal binding) as PMA-zeolite has increased surface area and adsorption capacity due to tribomechanical activation. It has a potential to bind bacterial toxins, heavy metals, and inflammatory mediators locally in the periodontal pockets.  This provides a biologically detoxifying effect, which standard mechanical cleaning lack. Moreover, anti-inflammatory effects have also been postulated for this material, potentially expected to reduce local inflammation. This may support tissue regeneration and healing in chronic periodontitis beyond what standard prophylaxis can achieve. At last, unlike antiseptics, PMA-zeolite does not indiscriminately kill bacteria and adsorbs pathogenic factors due to its physical-chemical properties in a selective manner without disrupting the entire oral microbiome. This is a property that was expected to promote long-term microbial balance. As a non-toxic material, PMA-zeolite avoids side effects of chemical antiseptics like mucosal irritation.

This explanation has now been added to the Introduction section.

  1. The repeated phrase “PMA-Zeolite treatment combined with standard prophylaxis” affects readability; consider using a consistent abbreviation after first mention.

This has now been abbreviated as PZ+SP (PMA-Zeolite plus Standard Prophylaxis).

  1. Strengthen the justification for the 12-week treatment period—why was this specific duration chosen, and how does it compare to similar studies?

According to the recent literature a minimum follow-up to evaluate the efficacy of nonsurgical periodontal therapy (NSPT) in treating periodontitis in patients with concurrent systemic conditions is 3 months. This is also time for recall, however evidence for a specific recall interval (e.g. every 3 months) for all patients following periodontal therapy depends on the various factors such as risk factors, type of periodontal disease and underlying systemic disease and is preferred to be individually determined. In our study, we cited study of Kaner D et al (2007) which indicated that administration of antimicrobial therapy immediately after initial mechanical debridement provides better clinical findings than late administration of systemic antimicrobial therapy with mechanical reinstrumentation after 3 months. In our cohort, based on this knowledge regarding the influence of time on the outcome of therapy, we determined a three-month treatment period and proved the effectiveness of the PMA-Zeolite therapy for 10 weeks on the decreased the gingival inflammation, decreased bleeding on probing (BoP) and pocket depth status (PD) parameters. We consider the period of monitoring the effect of zeolite therapy on periodontal changes to be optimal for 12 weeks in order to observe, assess and measure the therapeutic effect.

  1. The discussion of microbiome shifts is descriptive but could benefit from deeper interpretation—how do the observed taxonomic changes translate into clinical or mechanistic insights?

We discussed the observed changes in such a way not to overinterpret the obtained results. The possible mechanistical aspects have now been added in the discussion section as suggested.

(…) Moreover, PMA-Zeolite treatment combined with standard prophylaxis decreased the abundance of the family Desulfobulbaceae that is found to play a role in inflammation within the oral cavity of patients with periodontitis. This effect may be attributed to the adsorption of sulphur compounds by PMA-zeolite which may lead to decreased levels of these pro-inflammatory bacterial metabolites that support the growth of these taxa. Accordingly, a shift of the microbial community toward a more balanced state may then occur. (…)

(…) PMA-Zeolite treatment combined with standard prophylaxis also increased the abundance of the family Lachnospiraceae, that has dual roles in different pathogeneses but in connection with PMA-Zeolite treatment combined with standard prophylaxis may be evaluated as a potential signature for reduced histamine content in the oral cavity. This Lachnospiraceae potential to modulate histamine levels or maybe, outcompete histamine-producing taxa, may contribute to a less inflammatory environment.(…)

(…) PMA-Zeolite treatment combined with standard prophylaxis decreased the abundance of the genus Desulfobulbus that is found to play a role in inflammation within the oral cavity of patients with parodontosis [53-57]. Desulfobulbus oralis has previously been found to increase levels of IL-1α, IL-6 that activate the proinflammatory pathways (57). Both taxa may be suppressed by PMA-zeolite which leads to diminished inflammatory modulators. (…)

(…) At last, PMA-Zeolite treatment combined with standard prophylaxis decreased the abundance of the species Gemella bergeri that can become pathogenic and cause infec-tions in patients with poor oral hygiene [61, 62]. This reduction may result from the enhanced adsorptive properties of PMA-Zeolite, which can adsorb bacterial metabolites or even disrupt biofilm formation. This mechanism may impair the ecological niche for opportunistic pathogens like G. bergeri. (…)

(…) From obtained results, it may be deduced that PZ+SP has a modulatory effect on the oral microbiome through selective suppression of inflammation-associated taxa, i.e. Deltaproteobacteria, Desulfobulbaceae, Desulfobulbus oralis, and Gemella bergeri. These taxa have already been described in the literature as oral dysbiosis markers. The treatment also increased the abundance of genera Oribacterium and members of Lachnospiraceae that point to a shift toward a more anti-inflammatory microbial profile. These observations may be explained by the PMA-Zeolite’s ability to adsorb pro-inflammatory metabolites and interfere with biofilm formation while at the same time supporting microbial homeostasis. (…)

  1. The conclusion is overly concise; it should be expanded to highlight how the findings contribute to current knowledge, propose mechanisms, and outline specific directions for future research.

The conclusion has now been rewritten accordingly as follows:

(…) This study demonstrates that 10 weeks of treatment with PMA-Zeolite, when used as an adjunct to professional prophylaxis, offers significant clinical and microbiological benefits for the patients in the non-surgical treatment of periodontitis. The adjunctive application of PMA-Zeolite resulted in reduction in mucosal redness and clinical in-flammation, as reflected by decreased bleeding on probing (BoP), and a significant im-provement in periodontal pocket depth (PD). These findings suggest that PMA-Zeolite accelerates clinical improvements compared to standard prophylaxis alone. Given the increasing concerns about antibiotic resistance, PMA-Zeolite presents a promising non-antibiotic alternative or complementary treatment strategy in periodontal care.

Oral professional plaque removal combined with PMA-Zeolite treatment also re-duced some microbial markers linked with oral dysbiosis-inflammation, including Deltaproteobacteria, Desulfobulbaceae, Desulfobulbus, Desulfobulbus oralis, and Gemella bergeri while favoring the growth of Oribacterium along with Lachnospiraceae, which are associated with reduced histamine-related inflammation. These findings contribute to current knowledge by demonstrating that PMA-Zeolite can influence microbial composition in a way that supports periodontal health, through mechanisms including the adsorption of bacterial toxins, pro-inflammatory metabolites, and interference with pathogenic biofilm formation. The observed microbial shifts suggest that PMA-Zeolite promotes rebalancing of the oral microbiome, contributing to reduced inflammation and improved clinical outcomes.

These findings support the potential of PMA-Zeolite as an effective adjunct that can improve the outcomes of conventional nonsurgical periodontal therapy in patients with periodontitis. However, the efficacy of PMA-Zeolite must be confirmed through larger, long-term clinical trials involving broader patient populations. This study offers preliminary insight into its molecular effects but remains largely descriptive. Future research needs to explore its impact on cytokine signaling, immune regulation, and biofilm dynamics to clarify its full therapeutic potential. (…)

Reviewer 3 Report

Comments and Suggestions for Authors

The study should be changed for description of the results only, since it is a very small number of subjects and the results of efficacy cannot be taken from it. And even the conclusions are very limited.

Author Response

We acknowledge that the small sample size used in the presented study limits the statistical power and generalizability of the findings. This is why we state clearly in our conclusion that further studies are needed both for a more general assumptions and insight into the mechanisms. We agree that the study should be primarily interpreted as a descriptive analysis of microbiome and clinical trends, rather than a definitive evaluation of efficacy.

We stated this now clearly in the abstract and in the introduction.

We have also rewritten the conclusions to reflect the preliminary nature of the data and emphasize the need for larger-scale studies to validate these observations.

Round 2

Reviewer 2 Report

Comments and Suggestions for Authors

The authors responded to my comments properly. It can be published in the current version.

Author Response

Thank you for your comment. 

Reviewer 3 Report

Comments and Suggestions for Authors

All suggestions were done.

Author Response

Thank you for your comment.